# The Effects of Genistein at Different Concentrations on *MCF-7* Breast Cancer Cells and *BJ* Dermal Fibroblasts

**DOI:** 10.3390/ijms232012360

**Published:** 2022-10-15

**Authors:** Magda Aleksandra Pawlicka, Szymon Zmorzyński, Sylwia Popek-Marciniec, Agata Anna Filip

**Affiliations:** Department of Cancer Genetics with Cytogenetic Laboratory, Medical University of Lublin, Radziwiłłowska 11 Street, 20-080 Lublin, Poland

**Keywords:** genistein, breast cancer, *MCF-7* cells, skin, fibroblasts

## Abstract

This study aimed to evaluate the safety and potential use of soy isoflavones in the treatment of skin problems, difficult-to-heal wounds and postoperative scars in women after the oncological treatment of breast cancer. The effects of different concentrations of genistein as a representative of soy isoflavonoids on *MCF-7* tumor cells and *BJ* skin fibroblasts cultured in vitro were assessed. Genistein affects both healthy dermal *BJ* fibroblasts and cancerous *MCF-7* cells. The effect of the tested isoflavonoid is closely related to its concentration. High concentrations of genistein destroy *MCF-7* cancer cells, regardless of the exposure time, with a much greater effect on reducing cancer cell numbers at longer times (48 h). Lower concentrations of genistein (10 and 20 μM) increase the abundance of dermal fibroblasts. However, higher concentrations of genistein (50 μM and higher) are detrimental to fibroblasts at longer exposure times (48 h). Our studies indicate that although genistein shows high potential for use in the treatment of skin problems, wounds and surgical scars in women during and after breast cancer treatment, it is not completely safe. Introducing isoflavonoids to treatment requires further research into their mechanisms of action at the molecular level, taking into account genetic and immunological aspects. It is also necessary to conduct research in in vivo models, which will allow for eliminating adverse side effects of therapy.

## 1. Introduction

Breast cancer is a malignant tumor of the mammary gland originating from epithelial tissue and is also the most common malignant tumor in women [1]. In 2020, female breast cancer surpassed lung cancer as the most commonly diagnosed cancer, with an estimated 2.3 million new cases (11.7% of total cancers) [2]. Depending on the type of cancer, stage of the disease and patient’s condition, different treatment methods are applied [3,4]. The most used method is local or radical surgery supplemented with radiotherapy and/or chemotherapy [5]. There are also trials of novel treatments with fewer side effects, e.g., photodynamic therapy [6]. Although these methods show great promise, at this point they are rarely used due to the high cost of therapy. Modern medicine allows for breast reconstructive surgery, which can be performed during subtraction surgery or at another time [7]. However, regardless of the decision made by the surgeon and the patient, surgery is usually followed by an extensive scar. A complementary treatment used to eliminate the remaining cancer cells can also adversely affect postoperative wound healing and proper scar formation, which is not only an aesthetic defect but can also impede mobility and be painful. Therefore, it is essential to implement as soon as possible a therapy supporting these two processes [8]. Several preparations are available for the treatment of scars, but few of them have been sufficiently tested for safety in oncology patients. Many active substances can potentially stimulate ongoing tumorigenesis in tissues, but there is a lack of scar treatment products that would be safe for oncological patients but effective at the same time.

Soy isoflavonoids are classified as phytoestrogens, plant equivalents of female hormones [9,10]. Soy phytoestrogens, especially genistein and daidzein, in their structure and action resemble the female hormone estradiol [11]. Therefore, applied topically, they could bind to estrogen receptors in the skin and thus show a direct effect on the changes occurring in its cells [12,13]. The action of phytoestrogens on the skin and wound healing has been supported by numerous studies [14,15,16,17,18,19,20,21,22,23,24,25,26,27,28,29,30,31,32]. These compounds primarily affect the production of collagen, elastin and hyaluronic acid [23]. They also show strong soothing and anti-inflammatory properties. As flavonoid substances are characterized by strong anti-free solid radical properties, they also inhibit the activity of enzymes from the metalloproteinase class (MMP-1 and MMP-3), which are activated under the influence of UV radiation. Metalloproteinase MMP-1 breaks down collagen fibers, whereas MMP-3 breaks down the proteins responsible for proper communication between the epidermis and the dermis [33].

The aim of this study was to evaluate the safety and potential use of soy isoflavones in the treatment of skin problems, difficult-to-heal wounds and postoperative scars in women after oncological treatment of breast cancer. Therefore, a study was conducted to analyze the effects of different concentrations of genistein, as a representative of soy isoflavonoids, on *MCF-7* cancer cells and *BJ* dermal fibroblasts.

## 2. Results

Figure 1 and Figure 2 show the cytotoxic effects of genistein at different concentrations after 24 h and 48 h on tumor cells *MCF-7* and non-tumor cells *BJ* in MTT test. After 24 h of treatment, genistein was cytotoxic for *MCF-7* cells in concentrations over 80 μM and for *BJ* cells at its highest concentration (200 μM). It could also be observed that genistein at lower concentrations stimulated fibroblast growth. Treatment with genistein 50, 80, 100, 150 and 200 μM for 48 h, induced cytotoxicity in both non-tumor *BJ* and tumor *MCF-7* cells.

Figure 3 and Figure 4 show the influence of different concentrations of genistein after 24 h or 48 h on apoptosis of tumor *MCF-7* cells and non-tumor *BJ* cells. After 24 h, among *MCF-7* tumor line, apoptotic cells are observed at concentration as low as 20 μM. At genistein concentrations of 50 μM and higher, a reduction in the number of viable cells is observed. Among the cells of the non-tumor *BJ* line, apoptotic cells are observed only at a concentration of 150 μM. At lower concentrations of genistein, an increased number of viable fibroblasts can be observed. After 48 h incubation of *MCF-7* cells with genistein, apoptosis and a significant decrease in the number of viable cells is observed from a concentration of 20 μM. For this exposure time, the *BJ* fibroblast population is reduced already at genistein concentrations of 50 μM.

Figure 5 and Figure 6 show the impact of different concentrations of genistein on viability of tumor *MCF-7* cells and non-tumor *BJ* cells. A reduction in the number of *MCF-7* tumor cells above the genistein 20 μM concentration was observed at both exposure times. In the fibroblast population, at 24 h genistein exposure, a decrease in cell number was observed only at a concentration of 150 and 200 μM. When *BJ* line cells were incubated with genistein for 48 h, a significant decrease in population size was observed as low as 50 μM concentration.

Gene expression analysis showed that all genistein concentrations resulted in change of all gene expression (Figure 7). Genistein at a concentration of 20 μM also strongly stimulated the expression of the *MKI67* in the *MCF-7* line and mildly stimulated its expression in fibroblasts. The highest concentration of genistein negatively affected the expression of the *MKI67* in both tumor cells and dermal fibroblasts. Genistein at concentrations of 10 and 20 μM increased *BIRC5* gene expression in the *BJ* line, while at the highest concentration, it strongly decreased it. The lowest concentration of phytoestrogen also increased the expression of the *BIRC5* gene in the tumor line, while concentrations of 20 and 50 μM decreased it. The lowest concentration of genistein strongly stimulated *AKT1* expression in the *BJ* line, while the other concentrations resulted in decreased expression. All concentrations of genistein decreased *AKT1* expression, with a decreasing expression in proportion to increasing concentrations. It was observed that genistein increased *BCL2* expression in fibroblasts at all concentrations, with the strongest at a concentration of 10 μM. The lowest concentration of the phytoestrogen gently increased the expression of the analyzed gene, while the other concentrations caused a decrease.

## 3. Discussion

The tests performed clearly indicate that genistein affects both healthy dermal *BJ* fibroblasts and cancerous *MCF-7* cells. The effect of the tested isoflavonoid is closely related to its concentration. The authors’ study indicates that high concentrations of genistein destroy *MCF-7* cancer cells, regardless of the exposure time, with a much greater effect reduces cancer cell numbers at longer times (48 h). Lower concentrations of genistein (10 and 20 μM) increase the abundance of dermal fibroblasts. However, higher concentrations of genistein (50 μM and higher) prove to be detrimental to fibroblasts at longer exposure times (48 h).

The results of the *MCF-7* cell assay are in line with those of Kabała et al., who used two cancer lines, *MCF-7* and *MDA-MV-231* in their research. The conducted study showed a high dose-dependent effect in cell viability assays. The IC50 value was 47.5 μM for genistein. Isoflavonoid-induced apoptosis was dose- and time-dependent for both cell lines, with isoflavonoids being more active on *MCF-7* cells [34].

Choi also conducted similar studies et al. When *MCF-7* cells were treated with 50, 100, 150 and 200 μM of genistein for 24, 48 or 72 h, cell growth was significantly decreased in a concentration-dependent manner. After 48 h of treatment with 50 μM genistein, genistein-induced apoptosis features were observed in *MCF-7*. This observation is based on the finding that the expression of B-cell lymphoma protein 2 (Bcl-2) was decreased, whereas the expression of Bcl-2-related X protein (Bax) was induced by genistein. *BCL2* gene family includes genes associated with regulation of apoptosis; BCL2 protein itself acts as antiapoptotic factor, while BAX is proapoptotic one [34]. The results of their study suggested that the induction of apoptosis by genistein was associated with the modulation of estrogen receptor-α (ERα) and cyclin D1 expression, which resulted in the inhibition of *MCF-7* cells proliferation [35].

An important problem in the use of isoflavonoids in cancer therapy turns out to be the maximum physiological concentrations of the tested compounds. This problem was addressed by Tsuboy et al., who found that the maximum concentration of genistein that can be achieved in tissues is 30 μM [36]. Tsuboy et al. also determined that only supraphysiological levels of Gen (50 and 100 μM) were cytotoxic to these cell lines; concentrations of 10 and 25 μM did not induce apoptosis or significant changes in the expression of the genes. Positive results were found only in cell cycle analysis: G0/G1. Therefore, despite the lack of induction of apoptosis, genistein at physiologically relevant serum concentrations still exerts chemopreventive effects through cell cycle modulation [36].

The use of soy isoflavonoids for skin problems after cancer treatment therefore requires studies on lower concentrations of genistein. Uifălean at al. conducted such a study and showed that at relatively low concentrations (1.56–13.06 µM), genistein stimulated the cell growth of *MCF-7*, in contrast with control, while higher concentrations of this compound had an inhibitory effect. The inhibitory effects of genistein are due to various molecular mechanisms: high doses of isoflavones (typically > 20 µM Gen) have been shown to stimulate apoptosis, inhibit cell proliferation and survival, and have antioxidant and angiogenesis inhibitory effects in breast cancer cells. In the present authors’ study, the MTT assay indicated mild stimulation of tumor cell proliferation at a genistein concentration of 10 μM [37]. Therefore, the safest achievable systemic concentration of genistein that will have a beneficial effect on wound healing and scar formation after cancer treatment while at the same time not stimulating cancer cells may be 20 μM.

There are several papers in which authors have analyzed the molecular mechanisms and basic signaling pathways involved in both the stimulatory and inhibitory effects of genistein [37,38,39,40,41,42]. Lavigne et al. analyzed gene expression patterns in *MCF-7* line cells treated with physiological (1 and 5 μM) and pharmacological (25 μM) concentrations of genistein. Studies using oligonucleotide microarrays showed that genistein alters the expression of genes belonging to a broad spectrum of pathways, including the estrogen pathway and the p53 pathway. *TP53* gene encodes proteins that are important for DNA damage repair and apoptosis. At doses of 1 and 5 μM, genistein induced an expression pattern suggestive of increased mitogenic activity, confirming the proliferative response to genistein observed in the *MCF-7* cell cultures we conducted, while at a dose of 25 μM, genistein induced a pattern that likely contributes to increased apoptosis, decreased proliferation and reduced total cell number [38], which is also consistent with the cell culture results I obtained.

In their study, Sreenivasa et al. showed that genistein inhibits the growth of *MCF-7* cell line in a dose-dependent manner. The genistein-induced growth inhibition is accompanied by a reduction in the number of mitotic cells and overexpression of the cyclin-dependent kinase inhibitor p21WAF1, leading to cell cycle arrest. In addition, telomere length was significantly reduced in genistein-treated cancer cells. The analysis of a number of apoptosis-related genes showed the inhibition of Akt activity without affecting the level of Akt protein expression, as well as decreased expression of the pro-apoptotic gene *BAD*. The expression of one of the genes belonging to this group, *AKT1*, was also analyzed in our study. AKT1 is one of 3 closely related serine/threonine-protein kinases (AKT1, AKT2 and AKT3) called the AKT kinase, which regulate many processes including metabolism, proliferation, cell survival, growth, and angiogenesis. This is mediated through serine and/or threonine phosphorylation of a range of downstream substrates. The results of our study suggest a reduction in its expression at all tested genistein concentrations. Based on our own and other authors’ results, we can conclude that the inhibition of cell division by genistein is mediated in part by a decrease in telomere length, a decrease in mitotic divisions and inhibition of Akt activation, leading to the induction of apoptosis [39].

The antiproliferative effect of genistein on *MCF-7* cells and the molecular mechanisms involved in this effect were also analyzed by Prietsch et al. Their study showed that genistein induced phosphatidylserine externalization and LC3A/B immunopositivity in *MCF-7* cells, indicating apoptosis and cell death by autophagy. Genistein increased the proapoptotic BAX/Bcl-2 ratio threefold and induced a 20-fold reduction in anti-apoptotic survivin (BIRC-5). The reduced expression of the *BIRC-5* gene was also consistent with our analysis [40].

Chen et al. found that genistein inhibited *MCF-7* cell proliferation and induced cell apoptosis through the inactivation of IGF-1R and p-Akt and decreased the Bcl-2/Bax protein ratio. These results suggest that genistein inhibited cell proliferation by inactivating the IGF-1R-PI3 K/Akt pathway and reducing mRNA and protein expression of Bcl-2/Bax [41,42].

## 4. Materials and Methods

The *MCF-7* human breast cancer cells, media, trypsin and DMSO were purchased from the ATCC (Manassas, VA, USA), respectively, for use in the present study. *BJ* human skin fibroblasts were a gift from the Department of Synthesis and Chemical Technology of Medicinal Products in Medical University of Lublin. All plasticware used for cell culture were purchased from Sarsted (Nümbrecht, Germany). Insulin, Genistein, Trypan Blue Solution and Annexin V-Cy3TM Apoptosis Detection Kit were purchased from Sigma-Aldrich Co. (St. Louis, MO, USA). The Antibiotic-Antimycotic was purchased from Thermo-Fisher (Waltham, MA, USA). The Fetal Bovine Serum (FBS) was purchased form Biomed. The Cell Proliferation Kit (MTT) was purchased from Roche Diagnostics GmbH (Hong Kong, China), respectively. Genistein was dissolved in DMSO (final concentration of 0.1% in medium).

The *MCF-7* human breast cancer cells and *BJ* human skin fibroblasts were maintained in The Department of Cancer Genetics of Medical University in Lublin. The *MCF-7* cells were cultured in EMEM (Eagle’s medium), supplemented with 10% FBS, human recombinant insulin (0.01 mg/mL) and antibiotics (50 U/mL penicillin and 50 μg/mL streptomycin). The *BJ* cells were cultured in EMEM (Eagle’s medium), supplemented with 10% FBS and antibiotics (10,000 U/mL of penicillin, 10,000 µg/mL of streptomycin, and 25 µg/mL of Gibco Amphotericin B). Both of cultures were maintained at 37 °C in a humidified atmosphere containing 5% CO_2_.

Cytotoxicity was examined using MTT assay. Cells were plated at 2.5–5 × 10^5^ cells/well in a 96-well tissue culture plate and incubated for 24 h following which they were exposed to genistein solutions at concentrations of 10, 20, 50, 80, 100, 150 and 200 μM and DMSO solutions at concentration of 10% and 15%. Following incubation for 24 and 48 h, the plated cells were incubated with MTT (final concentration 0.5 mg/mL) for 4 h at 37 °C. Then the Solubilization solution was added into each well (100 μL). The plates were incubated overnight at 37 °C, 5% CO_2_, so that the complete dissolution of formazan was achieved. The absorbance of MTT formazan was determined at 570 nm using a microplate ELISA reader.

The apoptosis was detected using Annexin V-Cy3TM Apoptosis Detection Kit. Cells were plated at 0.5–1.0 × 10^6^ cells/ml on Petri dishes and incubated for 24 h following witch they were exposed to genistein solutions at concentrations of 10, 20, 50, 80 and 100 μM and DMSO solution at concentration of 10%. Following incubation for 24 h, cells were washed three times using binding buffer solution. Then, double label staining solution (AnnCy3 and 6-CFDA) was added to Petri dishes covered in aluminum foil. Cells were incubated for 10 min at room temperature and washed five times with binding buffer to remove excess label from the cells. Following that, the binding buffer was added to the cells and the dishes were covered with a coverslip and observed under a florescence microscope, using the correct filter and light source, and then photographed.

To quantify the percentage of cells that underwent apoptosis, a trypan blue assay was additionally performed. Cells were plated at 0.5–1.0 × 10^6^ cells/ml on Petri dishes and incubated for 24 h after which they were exposed to genistein solutions at concentrations of 10, 20, 50, 80 and 100 μM and DMSO solution at a of 10%. Following incubation for 24 h, the cells were detached from the plates with trypsin solution. An equal amount of trypan blue was added to the cell pellet and incubated for 2 min at room temperature. The suspension was then applied to a primary slide, covered with a coverslip and observed under the microscope. The number of dead stained cells per 100 total cells was then calculated.

Expression of *EGFR, BCL2, MKI7, BIRC5* and *AKT1* genes was measured using SYBR Green PCR Mastermix (Thermo Scientific), in duplicate in 96-well plates using the 7500 Real-Time PCR System from Applied Biosystems (Foster City, CA, USA). The results were then analyzed using the 7500 System software (Applied Biosystems). Each reaction was normalized to *GAPDH* reporter gene expression, and relative expression was calculated using the ΔΔCt method [43]. The sequences of the primers used are listed in Table 1.

## 5. Conclusions

Our studies indicate that despite the high potential of genistein for use in the treatment of skin problems, wounds, and surgical scars in women during and after breast cancer treatment, it cannot be recognized as totally safe. Soy isoflavonoids are substances with very high biological activity, so their introduction into treatment requires further studies on their mechanisms of action at the molecular level considering genetic and immunological aspects. It is also necessary to conduct studies in in vivo models. This will allow for the elimination of possible side effects due to adverse effects on the body or therapy.

## Figures and Tables

**Figure 1 ijms-23-12360-f001:**
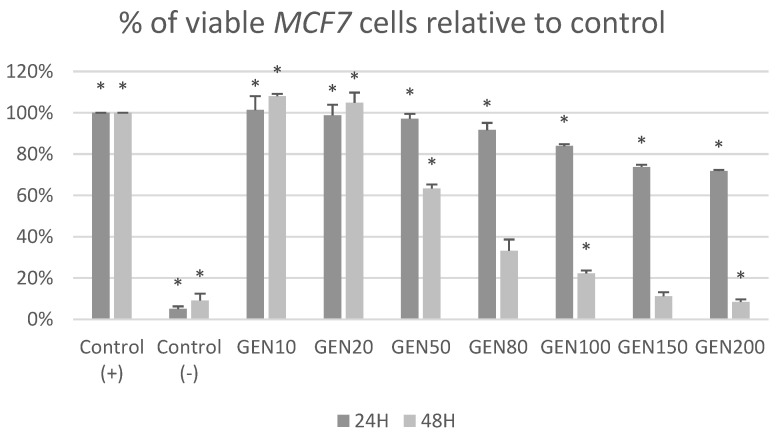
Cytotoxic effect of genistein at different concentrations after 24 h and 48 h on tumor *MCF-7* cells in MTT test. Results are the mean ± SD of n = 5. * Statistically significant (*p* < 0.05; ANOVA followed by Dunnett’s test compared with control group). Cells cultured with 0.1% DMSO were considered Control (+). Cells cultured with 15% DMSO were considered as Control (−).

**Figure 2 ijms-23-12360-f002:**
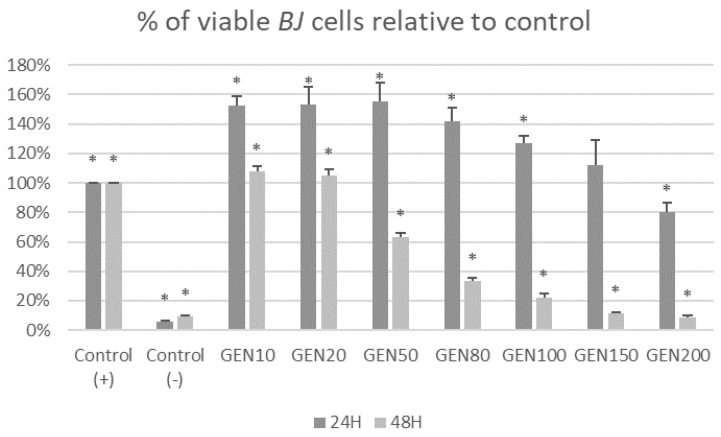
Cytotoxic effect of genistein at different concentrations after 24 h and 48 h on non-tumor *BJ* cells in MTT test. Results are the mean ± SD of n = 5. * Statistically significant (*p* < 0.05; ANOVA followed by Dunnett’s test compared with control group). Cells cultured with 0.1% DMSO were considered Control (+). Cells cultured with 15% DMSO were considered as Control (−).

**Figure 3 ijms-23-12360-f003:**
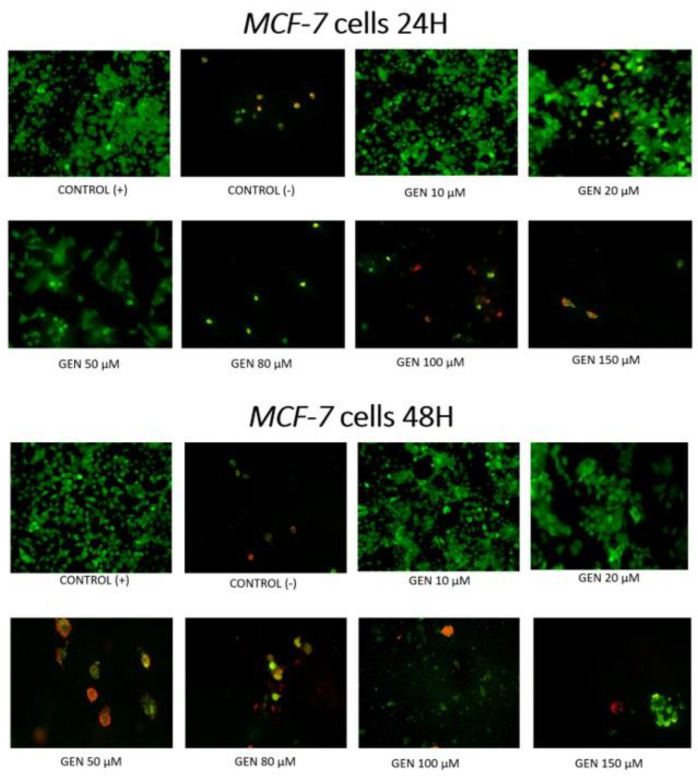
Induction of apoptosis by genistein at different concentrations in tumor *MCF-7* cell lines in fluorescent microscope (magnification 400×). Cells were stained with annexin-Cy3.18 (AnnCy3) and 6-Carboxyfluorescein diacetate (6-CFDA). AnnCy3 binds to phosphatidylserine present in the outer leaflet of plasma membrane of cells starting the apoptotic process. The binding is observed as red fluorescence. 6-CFDA is used to measure viability. When this non-fluorescent compound enters living cells, esterases hydrolyze it, producing fluorescent compound 6-carboxyfluorescein (6-CF). This appears as green fluorescence. Cells cultured with 0.1% DMSO were considered as Control (+). Cells cultured with 15% DMSO were considered as Control (−).

**Figure 4 ijms-23-12360-f004:**
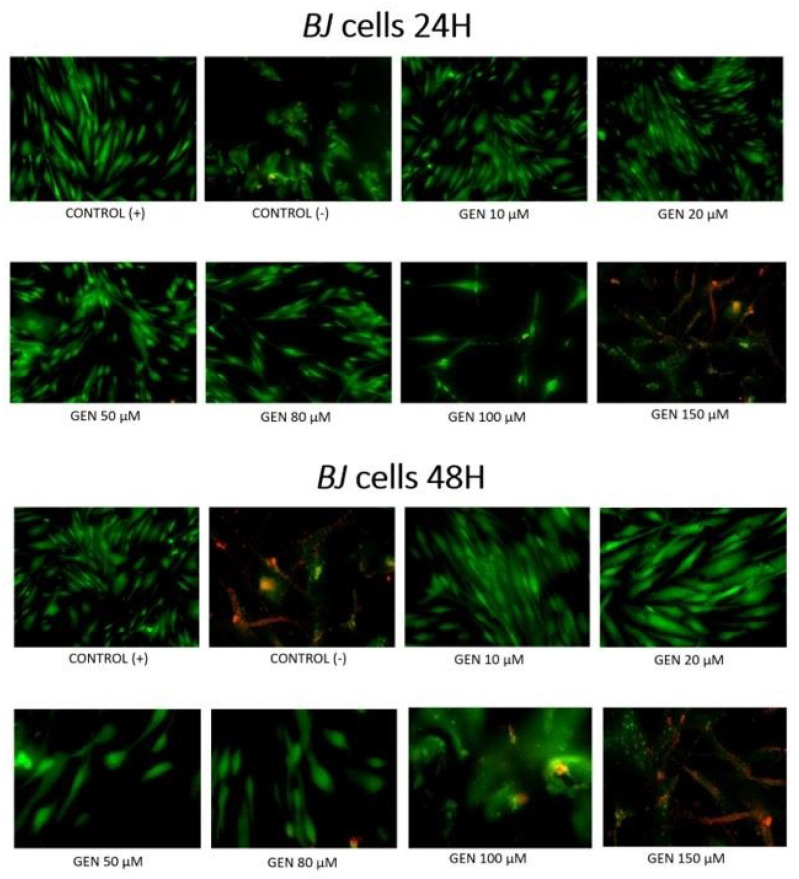
Induction of apoptosis by genistein at different concentrations in non-tumor *BJ* cell lines in fluorescent microscope (magnification 400×). Cells were stained with annexin-Cy3.18 (AnnCy3) and 6-Carboxyfluorescein diacetate (6-CFDA). AnnCy3 binds to phosphatidylserine present in the outer leaflet of plasma membrane of cells starting the apoptotic process. The binding is observed as red fluorescence. 6-CFDA is used to measure viability. When this non-fluorescent compound enters living cells, esterases hydrolyze it, producing fluorescent compound 6-carboxyfluorescein (6-CF). This appears as green fluorescence. Cells cultured with 0.1% DMSO were considered Control (+). Cells cultured with 15% DMSO were considered Control (−).

**Figure 5 ijms-23-12360-f005:**
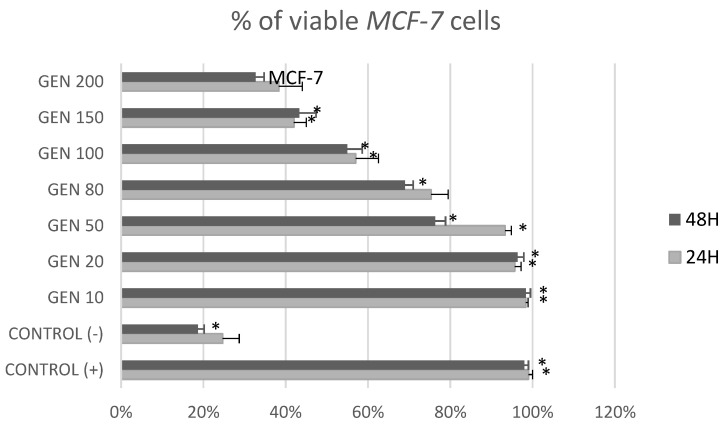
Percentage of viable MCF7 cells [%] in trypan blue exclusion assay after treatment with genistein at different concentration [μM]. Results are the mean ± SD of n = 3. * Statistically significant (*p* < 0.05; ANOVA followed by Dunnett’s test compared with control group). Cells cultured with 0.1% DMSO were considered Control (+). Cells cultured with 15% DMSO were considered Control (−).

**Figure 6 ijms-23-12360-f006:**
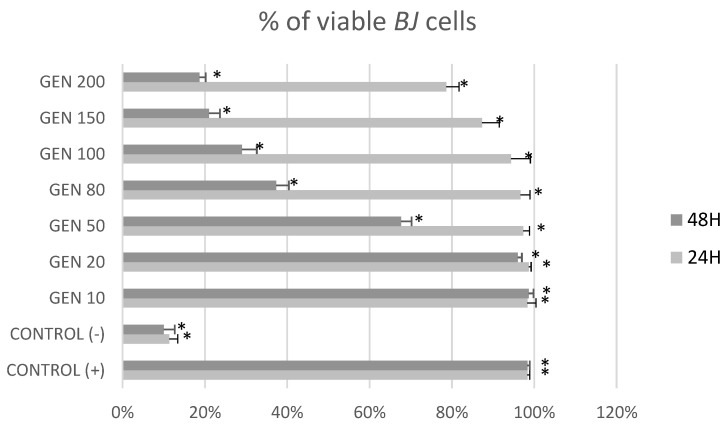
Percentage of viable *BJ* cells [%] in trypan blue exclusion assay after treatment with genistein at different concentration [μM]. Results are the mean ± SD of n = 3. * Statistically significant (*p* < 0.05; ANOVA followed by Dunnett’s test compared with control group). Cells cultured with 0.1% DMSO were considered Control (+). Cells cultured with 15% DMSO were considered Control (−).

**Figure 7 ijms-23-12360-f007:**
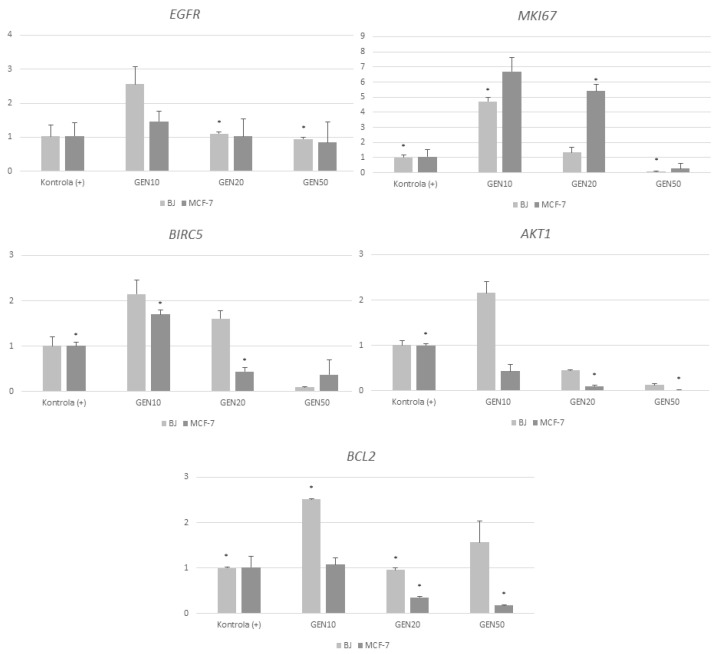
Analysis of gene expression in *MCF-7* and *BJ* cells treated with genistein at different concentrations 10, 20 and 50 μM) after 24 h Results are the mean value SD of n = 2. * Statistically significant (*p* < 0.05 ANOVA Friedman along with Dunn’s multiple comparisons test compared with control group). Cells cultured with 0,1% DMSO were considered as Control (+). Cells cultured with 15% DMSO were considered as Control (−).

**Table 1 ijms-23-12360-t001:** Sequences of primers.

Gene	Forward Sequence	Reverse Sequence	Product Size (bp)
*BCl2*	ATCGCCCTGTGGATGACTGAGT	GCCAGGAGAAATCAAACAGAGGC	140
*MKI67*	GAAAGAGTGGCAACCTGCCTTC	GCACCAAGTTTTACTACATCTGCC	151
*EGFR*	AACACCCTGGTCTGGAAGTACG	TCGTTGGACAGCCTTCAAGACC	106
*AKT1*	TGGACTACCTGCACTCGGAGAA	GTGCCGCAAAAGGTCTTCATGG	154
*BIRC5*	CCACTGAGAACGAGCCAGACTT	GTATTACAGGCGTAAGCCACCG	115

## Data Availability

All data supporting and reported results available at e-mail address magda.pawlicka1@gmail.com.

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
