# Peer review of "The Effects of Genistein at Different Concentrations on MCF-7 Breast Cancer Cells and BJ Dermal Fibroblasts"

_ijms, 2022, doi:10.3390/ijms232012360_

Round 1

Reviewer 1 Report

Fig 1 and 2 and 5 and 6 are the  same  result of MTT, merge them in one fig.

No molecular mechanism of apoptosis and cell proliferation.

Author Response

Dear Reviewer,
thank you very much for your detailed reviews. All the comments were very valuable to us. Please find the responses to each comment below. I hope that the manuscript become clearer and more eligible now.

Point 1: Fig 1 and 2 and 5 and 6 are the same result of MTT, merge them in one fig.

Response 1:  Figures 1 and 2 show results of MTT test and figures 5 and 6 show results of trypan blue exclusion assay. The trypan blue exclusion test was performed to quantify dead cells. We corrected descriptions of these figures.

Point 2: No molecular mechanism of apoptosis and cell proliferation.

Response 2: We expanded the article with results analyzing the expression of genes related to apoptosis, proliferation and angiogenesis.

Reviewer 2 Report

In the submitted paper "The effects of genistein at different concentrations on MCF-7 2 breast cancer cells and BJ dermal fibroblasts" the authors shown the effect on viability of Genistein on MCF7 cancer cells and BJ dermal fibroblas. The authors claims that Genistein display citotoxicity against MCF7 On the contrary, Genistein appear to stimulate growth of BJ cells at 24h.

Several issues need to be adressed before publication:

"High concentrations of genistein (above 20 15 μM) destroy MCF-7 cancer cells, regardless of the exposure time": this is not what the data shown, eg. Viability MCF7 50 uM 24h is around 100%.

Figure 1 vs Figure 5, could you explain the difference?

Figure 2 Vs Figure 6, could you explain the difference?

"Therefore, the safest achievable systemic concentration of 170 genistein, which will have a beneficial effect on the skin of people after cancer treatment, 171 wound healing processes and scar formation, and at the same time will not stimulate can-172 cer cells, turns out to be 20 μM."

This need to be re-phrased. No clinical data are presented, therefore no conclusion can be made.

In order to demonstrate that Gen stimulate growth of BJ cells, additional experiments need to be perform (eg. using differents concentrations of cells)

Author Response

Dear Reviewer,
thank you very much for your detailed review. All the comments were very valuable to us. Please find the responses to each comment below. I hope that the manuscript become clearer and more eligible now.

Point 1: High concentrations of genistein (above 20 15 μM) destroy MCF-7 cancer cells, regardless of the exposure time": this is not what the data shown, eg. Viability MCF7 50 uM 24h is around 100%.

Response 1: This sentence was meant to refer to longer exposure to genistein at
concentrations above 20 uM. It was not clear, so we rephrased the sentence.

Point 2:  Figure 1 vs Figure 5, could you explain the difference? Figure 2 Vs Figure 6, could you explain the difference?

Response 2: Figures 1 and 2 show results of MTT test and figures 5 and 6 show results of trypan blue exclusion assay. The trypan blue exclusion test was performed to quantify dead cells. We corrected descriptions of these figures.

Point 3: Therefore, the safest achievable systemic concentration of genistein, which will have a beneficial effect on the skin of people after cancer treatment, wound healing processes and scar formation, and at the same time will not stimulate cancer cells, turns out to be 20 μM. This need to be re-phrased. No clinical data are presented, therefore no conclusion can be made.

Response 3: Good point, we rephrased this sentence.

Point 4: In order to demonstrate that Gen stimulate growth of BJ cells, additional experiments need to be perform (eg. using differents concentrations of cells)

Response 4:  We also conducted analysis of expression of genes associated with cell proliferation on both cell lines. We have enriched the article with these results.

Reviewer 3 Report

The authors demonstrated beneficial “The effects of genistein at different concentrations on MCF-7 2 breast cancer cells and BJ dermal fibroblasts” in various areas. More details are needed as follows;

1-      The English structure showed to be improved a lot.

I have attached a corrected file with some suggestions as a model for corrections.

2-      In the introduction, the idea that cancer treatment methods should be reviewed in detail. The laser treatment showed be included. For Example;  Gamal, Hend, Walid Tawfik, Heba Mohamed Fahmy, and Hassan H. El-Sayyad. "Breakthroughs of using Photodynamic Therapy and Gold Nanoparticles in Cancer Treatment." IEEE Xplore, IEEE International Conference on Nanoelectronics, Nanophotonics, Nanomaterials, Nanobioscience & Nanotechnology (5NANO), pp. 1-4. IEEE, 2021. DOI: 10.1109/5NANO51638.2021.9491133

Author Response

Dear Reviewer,
thank you very much for your detailed review. All the comments were very valuable to us. Please find the responses to each comment below. I hope that the manuscript become clearer and more eligible now.

Point 1: The English structure showed to be improved a lot.

Response 1: Thank you for any corrections. We applied it. The paper was reviewed as well.

Point 2: In the introduction, the idea that cancer treatment methods should be reviewed in detail. The laser treatment showed be included. For Example;  Gamal, Hend, Walid Tawfik, Heba Mohamed Fahmy, and Hassan H. El-Sayyad. "Breakthroughs of using Photodynamic Therapy and Gold Nanoparticles in Cancer Treatment." IEEE Xplore, IEEE International Conference on Nanoelectronics, Nanophotonics, Nanomaterials, Nanobioscience &
Nanotechnology (5NANO), pp. 1-4. IEEE, 2021. DOI: 10.1109/5NANO51638.2021.9491133

Response 2: In the Introduction we mentioned modern breast cancer treatment methods, including Photodynamic Therapy. 

Best regards
Magda Pawlicka

Round 2

Reviewer 1 Report

Reject.

Reviewer 2 Report

Na